# The Coming of Age of the P2X7 Receptor in Diagnostic Medicine

**DOI:** 10.3390/ijms24119465

**Published:** 2023-05-30

**Authors:** Francesco Di Virgilio, Valentina Vultaggio-Poma, Simonetta Falzoni, Anna Lisa Giuliani

**Affiliations:** Department of Medical Sciences, University of Ferrara, 44121 Ferrara, Italy; vltvnt@unife.it (V.V.-P.); simonetta.falzoni@unife.it (S.F.); annalisa.giuliani@unife.it (A.L.G.)

**Keywords:** P2X7, PET, neuroinflammation, biomarker, acute-phase reactant

## Abstract

The discovery of the P2X7 receptor (P2X7R, originally named P2Z) in immune cells, its cloning, and the identification of its role in a multiplicity of immune-mediated diseases raised great hopes for the development of novel and more potent anti-inflammatory medicaments. Unfortunately, such hopes were partially deluded by the unsatisfactory results of most early clinical trials. This failure substantially reduced the interest of the pharmaceutical and biotech industries in the clinical development of P2X7R-targeted therapies. However, recent findings ushered in a second life for the P2X7R in diagnostic medicine. New P2X7R radioligands proved to be very reliable tools for the diagnosis of neuroinflammation in preclinical and clinical studies, and detection and measurement of free P2X7 receptor (or P2X7 subunit) in human blood suggested its potential use as a circulating marker of inflammation. Here we provide a brief review of these novel developments.

## 1. Introduction

The global anti-inflammatory drug market is steadily increasing, rising from about 99 billion USD in 2021 to over 127 billion USD in 2030, but according to some estimates, it could amount to even 191.42 billion USD as early as 2027 (https://www.fortunebusinessinsights.com/anti-inflammatory-drugs-market-102825, accessed on 17 April 2023; https://www.precedenceresearch.com/anti-inflammatory-therapeutics-market, accessed on 17 April 2023). Nevertheless, anti-inflammatory drug classes, despite the large number of commercial specialties, are few: non-steroid anti-inflammatory drugs (NSAIDs), steroidal anti-inflammatory drugs (SAIDs), anti-histamine drugs, and immunosuppressants. Therefore, there is an increasing and urgent need not just for novel drugs but, more importantly, for novel classes of compounds targeting novel molecular targets. The cloning of the P2X7 receptor (P2X7R, at the time known as P2Z) [1], in the wake of a cogent series of in vitro experiments showing its likely involvement in inflammation [2,3,4,5], spurred great interest and hopes that identification of this novel target would pave the way to new, more effective anti-inflammatory drugs. However, Phase I/II clinical trials launched by major pharmaceutical companies to explore the effect of P2X7R antagonism in several chronic inflammatory diseases (rheumatoid arthritis, inflammatory bowel disease, depression, osteoarthritis, chronic obstructive pulmonary disease, and Crohn’s disease) produced rather disappointing results [6,7,8,9], which all together cooled down interest in P2X7R as a suitable target for anti-inflammatory therapy. However, despite diffuse skepticism, a few biotech and pharmaceutical industries kept alive their interest in this receptor and persisted in the development of P2X7R-targeting compounds (e.g., Johnson and Johnson for bipolar disorders, Evotec for rheumatoid arthritis and other chronic inflammatory diseases, RaqualiaPharma/Asahi Kasei Pharma for neuropathic pain; see https://www.pharmaceutical-technology.com/data-insights/p2x7-antagonist-johnson-johnson-bipolar-disorder-manic-depression-likelihood-of-approval/; https://www.alliedmarketresearch.com/p2x7-receptor-antagonists-market-A10362; https://investor.lilly.com/news-releases/news-release-details/lilly-and-asahi-kasei-pharma-announce-license-agreement-chronic; https://adisinsight.springer.com/drugs/800039286; https://www.pharmaceutical-technology.com/data-analysis/ak-1780-what-is-the-likelihood-that-drug-will-be-approved/, accessed on 17 April 2023).

Some unconventional further indications for a P2X7R-targeted therapy have also been proposed, for example, in the treatment of age-related retinal degeneration (AMD) (US patent n. US 2016/0263114A1, Inflammasome Therapeutics, https://www.inflam.com/, accessed on 17 April 2023) [10,11]. The suitability of P2X7R-targeting to treat AMD is also under scrutiny by the Danish company Breye Therapeutics (https://breye.com/pipeline/, accessed on 17 April 2023), and in addition, P2X7R has been proposed as a target for the cure of diabetic retinopathy [12,13] or glaucoma [14,15]. However, to our knowledge, no clinical trials have been started so far to investigate the efficacy of P2X7R targeting in eye diseases.

The high expression level of P2X7R in tumors has inspired therapeutic strategies based on the development of P2X7R-specific antibodies to be exploited in immunotherapy of cancer. This approach is based on the original observation that some tumors express a peculiar, non-functional variant of this receptor (referred to as “non-functional” P2X7, nfP2X7) that might be a tumor-specific antigen susceptible to immune targeting [16,17]. An open-label Phase I clinical trial in patients affected by basal cell carcinoma showed that topical administration of an ointment containing sheep anti-nfP2X7 polyclonal antibodies caused a statistically significant reduction of cancer lesions in 65% of patients [18]. Biosceptre Ltd., a Cambridge (UK)-based biopharmaceutical company, is currently developing a nfP2X7-targeted chimeric antigen receptor (CAR)-T cell therapy to treat different cancer types (https://www.biosceptre.com/innovation/nfp2x7/, accessed on 17 April 2023).

In the meantime, novel indications for P2X7R in clinical medicine have started surfacing based on the assumption that because P2X7R is highly expressed by inflammatory cells, inflammatory lesions richly infiltrated by inflammatory cells should show increased binding of P2X7R ligands. This hypothesis prompted the synthesis of P2X7R antagonist-based radioligands for the diagnosis of neuroinflammatory and neurodegenerative diseases [19,20]. However, radiodiagnosis is not the only “second life” of the P2X7R as a target in clinical diagnosis. As initially demonstrated by our laboratory [21] and later confirmed by Engel and co-workers [22] and Pelegrin and co-workers [23], the P2X7R, whether as the trimeric oligomer or as the P2X7 monomeric subunit, can be identified in the plasma or serum fractions of human blood, where its concentration increases during infection and systemic inflammation. This has led us and others to propose that the soluble or shed P2X7R or P2X7 subunit (sP2X7) might be additional useful circulating biomarker of inflammation (Figure 1).

## 2. The P2X7 Receptor as a Target for Positron Emission Tomography (PET)

Ever since its introduction for the diagnosis of brain pathologies, positron emission tomography (PET) has enjoyed wide popularity, its high cost withstanding, for the ability to image areas of the body suffering from abnormal patterns of blood flow or metabolism or affected by inflammatory processes. Thus, this technique has become a routine radio-diagnostic approach for the diagnosis of cancer and inflammation [24,25], as well as to understand the neurochemical basis of psychiatric disorders [20]. The increasing reliability of PET scans for the identification of disease areas of the body prompted the search for suitable tissue targets that were overexpressed under different disease conditions and thus possibly specifically and selectively bound by the radiotracers. The availability of reliable “radio targets” can obviously provide efficient anatomical localization of disease processes and make therapeutic approaches more specific and efficient.

The search for useful targets for PET-based diagnosis was very active in neurology and psychiatry, aimed at identifying abnormal circulatory or metabolic patterns that might underlie neurologic or psychiatric diseases [20]. Interestingly, often radiotracers were developed starting from pharmacologically well-characterized drug-like compounds that failed as therapeutics in clinical trials [20]. Not surprisingly, this is also the path that led to the development of P2X7R-targeted radiotracers.

The first evidence that the P2X7R could be a suitable target for PET imaging was provided by Zheng and co-workers who labeled with ^11^C the Glaxo GSK1482160 compound [26], soon followed by Bormans and co-workers who ^11^C-radiolabelled the Johnson and Johnson (J&J) JNJ-54173717 compound [27]. Several ^18^F-labelled antagonists were later synthesized, e.g., [^18^F]-EFB, developed from the Abbott antagonist A-804598 [28], the GSK1482160 derivative [^18^F]-IUR-1601 [29,30], the J&J compound [^18^F]-JNJ-64413739 [31], and the Janssen compound [^18^F]-PTTP [32] (Table 1).

Fluorine-labeled derivatives are in principle better than carbon-labeled compounds due to their longer half-lives. As of March 2023, about 24 PubMed studies, the first published in 2015, reported the synthesis and in vitro or in vivo investigation of over 10 different radiotracers for imaging the P2X7R in the brain [34]. One of these studies also reported an application outside the central nervous system (CNS) [32]. Of interest, first-in-man studies have already been performed with very positive results in healthy volunteers, Parkinson’s disease (PD), and multiple sclerosis (MS) patients, where the [^11^C]-JNJ-54173717 or the [^11^C]-SMW139 radioligands allowed quantitative analysis of P2X7R expression in the brain [33,35,36]. In regard to disease discrimination, no major differences in [^11^C]-JNJ54173717 radiotracer uptake were observed in healthy versus PD subjects in the study by Van Weehaeghe et al. [35], while in the study by Hagens et al. [33], [^11^C]-SMW139 accumulated at a higher level at sites of inflammatory lesions in patients affected by relapsing-remitting MS. Recently, the [^18^F]-JNJ-64413739 radiotracer has also been applied to epilepsy studies, showing a higher uptake in the brain and peripheral organs of kainic acid-injected mice (to induce status epilepticus) [37]. Increased radioligand uptake was also found in ex vivo brain slices from patients with drug-refractory temporal lobe epilepsy [37]. A caveat was raised about the possible interference with radioligand binding of the loss-of-function P2X7 SNP rs3751143 [35]. However, given the low frequency (less than 2%) of this SNP in homozygosity [38], this is unlikely to significantly affect P2X7R-targeted radioligand uptake under most clinical conditions. Thus, we may tentatively conclude that, based on these preliminary but convergent pre-clinical and clinical studies, the P2X7R will have a bright future in the radiodiagnosis of neuroinflammation, also due to the current lack of fully satisfactory PET ligands.

The most commonly used target in the diagnosis of neuroinflammation has traditionally been the translocator protein 18 kDa (TSPO), a protein localized to the outer mitochondrial membrane involved in cholesterol transport [39,40]. However, first-generation ligands suffered from low brain uptake and high non-specific binding and were affected by the presence of TSPO SNPs that caused large variations in binding affinity [41]. Such drawbacks now seem to have been overcome with the introduction of second-generation agents characterized by high affinity and selectivity, high brain uptake, and metabolic stability [20]. Recently, a novel ligand that seems to be insensitive to the major SNP affecting radioligand binding to TSPO has also been described [42]. The rationale for using TSPO is its association with M1 microglial cell polarization and, therefore, its overexpression at sites of inflammation in the brain [43]. However, it should be remembered that microglia are a very plastic cell type that can transition from one differentiation state to another in the context of neuroinflammation and in different areas of the brain; thus, identification of neuroinflammation exclusively based on the M1 microglia phenotype could be reductive [44]. In addition to TSPO, other targets are also being investigated for PET diagnosis: the metabolic glutamate receptor 5, cyclooxygenase (COX)-1, COX-2, inducible nitric oxide synthase (iNOS), colony-stimulating factor-1 (CSF1) receptor, sphingosine-1-phosphate receptor 1 (S1PR1), the histamine H1 receptor, the histamine H3 receptor, the α7-nicotinic acetylcholine receptor, phosphodiesterase 10A, the nociceptin/orphanin FQ peptide receptor, the synaptic vesicle protein 2A, histone deacytilases, type-1 cannabinoid receptor, and fatty acid amide dehydrolase (see [20,41,45] for recent reviews). However, while very promising, all these targets still suffer from numerous drawbacks that challenge their successful adoption for the clinical diagnosis of inflammatory diseases inside or outside the CNS.

In this regard, the development of P2X7R radioligands is in principle also useful in anti-cancer therapy, as according to the study by Fu et al. [32], the [^18^F]-PTTP ligand might be suitable to discriminate cancer (ectopic lung cell carcinoma) from inflammatory lesions due to its lower uptake by the tumor tissue compared to an inflammatory site. Peak uptake of [^18^F]-PTTP at the inflammatory site occurred about 5 min after injection of the probe and was about twice as high as uptake recorded in the tumor tissue. In the inflammatory site but not in the tumor, [^18^F]-PTTP levels declined during the following 60 min. The authors ascribe higher radioligand uptake at inflammatory sites to the higher infiltration by macrophages, a cell type well known to express the P2X7R at a high level, but the reason why uptake declines during the following several minutes is not clear [32]. Independent of the ability to discriminate tumors from inflammatory lesions, P2X7R-targeted radioligands might well support [^18^F]-deoxyglucose in the diagnosis of extra-CNS inflammatory diseases, such as atherosclerosis or cardiac amyloidosis [46].

Ideally, to be useful for the diagnosis of CNS diseases, the novel P2X7R radioligands should have high permeability across tissues and be able to cross the blood-brain barrier. In addition, high-affinity ligands are desirable to allow P2X7R binding in an environment, such as the inflammatory or tumor microenvironments, experiencing high extracellular ATP (eATP) levels [47,48]. This is especially relevant for the radioligands based on orthosteric antagonists, while those based on allosteric inhibitors (e.g., GSK1482160) might be less affected by high eATP levels. Availability of P2X7R-targeted radioligands might be beneficial for cancer therapy; in fact, although most cancers so far investigated show high P2X7R expression levels [47,49,50], this is likely not to be true of every tumor (for example, see [51,52,53]). Thus, it might be useful to devise a diagnostic technique that, based on the level of P2X7R expression in any given tumor, is able to provide a robust indication as to whether a P2X7R-targeted therapy is advisable. Such a technique might also be useful to verify the therapeutic efficacy of P2X7R-targeted antagonists, as it is anticipated that an efficient P2X7R-targeted therapy should displace the bound radioligand or decrease its uptake by the target tissue.

A challenge towards the development of P2X7R PET radioligands might be the expression by several cancers of the recently identified nfP2X7R, which might bind currently available radioligands with low affinity [16,17,54]. On the same line, another relevant issue for the translation of preclinical studies to the clinic is the different activity at the rodent or human receptor, respectively, that some of the radioligands may exhibit [28]. Another hurdle is, in principle, the presence of P2X7R polymorphisms that may affect P2X7R binding [55,56], but so far there is no published indication that the most common known P2X7R SNPs may affect antagonist binding. Thus, it is fair to conclude that the P2X7R is an appealing target for the development of novel and more specific radioligands for the investigation of inflammation and cancer.

## 3. The Shed P2X7 Receptor (or P2X7 Subunit) as a Circulating Marker of Inflammation

Inflammation causes several well-known systemic changes such as fever, leukocytosis (or sometimes leukopenia), metabolic derangements, circulatory and endocrine changes, and the release into the circulation of various molecules referred to as “acute-phase proteins”, the best known and generally used in clinical diagnosis being C-reactive protein (CRP) [57]. However, currently used acute-phase reactants are mostly highly non-specific as changes in their blood levels occur in responses to diverse inflammatory conditions, both local (when intense) and systemic, with the relevant exception of pro-calcitonin (PCT), irrespective of whether caused by pathogens (septic inflammation) or endogenous triggers (sterile inflammation). Furthermore, although changes in concentration of acute phase proteins also occur in association with chronic inflammation (e.g., increases in CRP during autoimmune or autoinflammatory diseases), most relevant changes occur in association with acute inflammation [58]. Thus, the identification of additional biomarkers of inflammation would be helpful for diagnostic and therapeutic purposes. In principle, the P2X7R is an appealing candidate due to its key role in the stimulation of the NLRP3 inflammasome and promotion of the first phases of inflammation via the release of IL-1β and IL-18 [59,60]. Contrary to most other acute phase reactants, the P2X7R is an integral plasma membrane protein; therefore, it is not obviously shed into the extracellular space, although there are important precedents, e.g., soluble TNF [61], IL-1β [62], or IL-2 [63] receptors, that have, however, never reached routine clinical diagnostic status. Scattered reports have documented the presence of low levels of the P2X7R in the serum or plasma of healthy subjects (average concentrations ranging from about 15 to 190 ng/L, depending on the laboratory where the determination was carried out [21,22,23,64]) (Table 2).

Given that the P2X7R is a homo-trimeric complex and that the ELISA kit currently available detects the individual homomer or fragments of it, it is uncertain whether the full trimeric receptor (possibly within extracellular vesicles or microparticles, MPs) or its individual monomers are released into circulation. Furthermore, it is still too early to identify the optimal blood fraction (plasma or serum) for its accurate determination. Pelegrin and coworkers [23] and Engel and coworkers [22] used plasma samples, while Giuliani et al. [21] tested the shed P2X7R (sP2X7R) concentration in both plasma and serum samples without finding substantial differences between the two sources. In all these studies, a good correlation was found between sP2X7R and CRP plasma/serum levels. Of relevance, Giuliani et al. found a very good correlation between CRP and sP2X7R levels in patients with brain ischemia, while the correlation was less stringent in patients with cancer [21]. Engel and co-workers investigated the blood sP2X7R concentration in subjects with temporal lobe epilepsy (TLE) or with psychogenic non-epileptic seizures (PNES), finding statistically significant increased sP2X7R levels in patients with TLE versus both healthy and PNES subjects [22]. Receiver Operating Characteristic (ROC) analysis showed good sensitivity and specificity of sP2X7R measurements in the differentiation of TLE subjects from both healthy and PNES subjects [22]. In their investigation of sP2X7R blood levels, Pelegrin and co-workers correlated sP2X7R with COVID-19 progression and with blood levels of key cytokines involved in the pathogenesis of this disease [23]. Quite interestingly, the increase in sP2X7R blood concentration correlated in a statistically significant fashion with disease severity, with the presence of severe COVID-19 symptoms, and with IL-18 levels.

The paucity of studies does not yet allow for reliable reference values for the sP2X7R concentration in healthy subjects. All the groups responsible for the reported measurements used an ELISA kit (the only one currently available) from the same manufacturer. Pelegrin and co-workers [23] and Engel and co-workers [22] measured P2X7R levels in plasma, while Giuliani and co-workers mainly used serum [21]. However, these latter investigators also ran parallel determinations in plasma and serum from the very same subjects without finding substantial differences. Thus, while the reason for the large variability reported in healthy subjects, from about 15 to 190 ng/L, is not clear, it is also clear that better standardized analytical techniques and larger subject cohorts are needed. In addition, although there is evidence that sP2X7R concentration in the blood may increase with age, no systematic studies have yet been carried out, nor has the effect of gender been investigated. A further complication might be the association of sP2X7R in the blood with circulating MPs. Several years ago, it was observed that MPs shed from human dendritic cells bear an ATP-responsive P2X7R on the membrane [65], and more recently, we found that about 20% of immunoreactive sP2X7R in the blood was associated with the MP fraction [21]. We think that this might be an underestimation of the actual contribution of MPs to P2X7R blood levels since MPs are very fragile and a substantial amount might break during the isolation procedure. Another potential factor affecting sP2X7R determination in blood is the presence of platelets. Although some studies showed a negligible presence of the P2X7R on human platelets [66], others, ours included, have reported a substantial level of expression of this receptor on platelets [21,67]. This finding, depending on the level of contamination by platelets, might bias the measurement of sP2RX7 in whole plasma. The level of contamination by platelets also depends on the total blood platelet concentration, a highly variable hematologic parameter with normal values ranging from 150 to 400 × 10^9^/L. Interestingly, at variance with the large variability in P2X7R levels reported in healthy subjects by the different laboratories, average values measured in patients, whether affected by COVID-19, epilepsy, or a miscellanea of inflammatory diseases, turn out to be very similar (Table 2).

As mentioned above, it has not been possible to verify whether the “shed P2X7R form” is the trimeric receptor complex, the full-length protein, or a cleaved fragment since epitopes bound by the antibodies used in the commercial kit are not known. We were unable to identify by Western blot the P2X7 subunit in serum or plasma due to the large number of serum/plasma proteins running in the 60–80 kDa region [21]. However, since the MP-derived P2X7 subunit runs in denaturing PAGE with the anticipated molecular mass (e.g., in the 72–75 kDa range) and since it is likely that MP-associated P2X7R contributes to the overall P2X7R blood levels, we believe that sP2X7 is most likely the whole-length protein. This implies that, since the P2X7 subunit is an integral membrane protein, it is also likely that MP shedding is the main route for release, as it is difficult to imagine how an integral membrane protein could be released without being cleaved. Alternatively, a cleaved P2X7 form could also be released into circulation, as described by Gorecki and co-workers, who identified a 35 kDa proteolytic fragment in Bz-ATP-stimulated cells incubated either in the presence of matrix metalloproteinase 2 (MMP-2) or under conditions that promote MMP-2 release [68]. Since MMP-2 is present and is further accumulated in the blood following P2X7R activation by eATP (or Bz-ATP), it is possible that MP-associated P2X7R is a substrate for MMP-2 activity, thus generating low-MW fragments. In any case, we were unable to detect P2X7 cleavage fragments in our studies due to a lack of specific antibodies and proper detection methods (i.e., an ELISA kit). At inflammatory sites, the contextual stimulation by eATP of P2X7R and MMP-2 release, both due to P2X7R activation, might trigger an auto-amplification loop ending with the accumulation of sP2X7R and cleavage fragments thereof. Thus far, blood has been the only biological fluid in which sP2X7R has been detected. Verification of whether, under disease conditions, this receptor also accumulates in other fluids would be of the utmost interest for the diagnosis of localized inflammation.

## 4. P2X7R and COVID-19

Severely ill COVID-19 patients develop a cytokine storm syndrome causing severe pneumonia and thrombo-inflammation, eventually causing intractable multi-organ failure (MOF) [69]. Cytokine storms are characterized by diffuse activation of innate immune cells (reminiscent of the macrophage-activation syndrome typical of some rheumatological diseases [70]), exhaustion of T lymphocytes, and unrestricted release of inflammatory (but also anti-inflammatory) cytokines. The P2X7R is the unique target of eATP, the prototypical damage-associated molecular pattern (DAMP) released at all inflammatory or damage sites, and a potent stimulant of cytokine release [55,71], which is in principle always activated during infection and inflammation. Furthermore, during COVID-19, systemic conditions are generated that are well known to facilitate P2X7R activation even at relatively low doses of eATP, for example, hypomagnesemia [72] or accumulation in the blood of the anti-microbial peptide LL-37 [73]. Altogether, this evidence has spurred speculation on the participation of the P2X7R in the pathogenesis of COVID-19 [74,75].

The level of circulating sP2X7R might be a reliable and easy-to-measure readout of the extent of P2X7R engagement during COVID-19. In fact, its activation by eATP is by itself a powerful stimulus for receptor shedding via an MP-mediated mechanism [21]. This mechanism has likely an important function in the overall homeostasis of the inflammatory response because, on the one hand, it down-modulates innate immune cell responses by removing from the plasma membrane the P2X7R as a potent stimulatory receptor (thus, for example, inhibiting NLRP3 inflammasome activation), and on the other, it facilitates the spreading of MP-carried pro-inflammatory factors in the circulation. Anticipations of an important role for the P2X7R in COVID-19 pathogenesis and possibly in its diagnosis and prognosis were confirmed by the recent investigation carried out in our and Pablo Pelegrin’s laboratories [23,76]. Both groups showed that sP2X7R circulating levels were increased in COVID-19 patients and correlated with disease severity. In the study by Pelegrin and co-workers, a good correlation was found between sP2X7R, CRP, and IL-18 levels, while a correlation between sP2X7R and PCT was found only in severely ill patients [23]. In the study by Giuliani et al., sP2X7R showed no correlation with CRP levels but a very good correlation with PCT [76]. A relevant finding from this latter study was the identification of sP2X7R as the unique analyte able to predict disease outcome at hospital admission [76]. Another intriguing observation was the strong positive correlation between sP2X7R and the anti-inflammatory cytokine IL-10. This raises the issue of whether the function of IL-10 in COVID-19 should be re-evaluated, underlying its role not just as a feed-back factor dampening inflammation but rather as a bona-fide pro-inflammatory mediator with a crucial function in disease spreading and amplification [76].

An ancillary relevant issue concerns the mechanism responsible for the sP2X7R release. Pelegrin and co-workers suggested that a main mechanism for sP2X7R release was via inflammasome activation, whether the canonical, P2X7R-stimulated NLRP3 or the P2X7R-independent pyrine inflammasome [23]. On the contrary, we think that sP2X7R is released simply in response to plasma membrane expressed P2X7R stimulation since no NLRP3 priming is necessary to trigger receptor shedding in our experimental models [21,65,76]. Our conclusion is also strengthened by the dissociation between sP2X7R, NLRP3, and IL-1β blood levels [76], which again suggests that the P2X7R is shed independently of inflammasome activation. Of course, should this mechanism be operative in vivo, the obvious implication is that sufficient amounts of eATP should be present in the blood. Anecdotal reports suggest that this might be the case in COVID-19 [77,78].

## 5. Conclusions and Further Developments

The transition to clinics for P2X7R-targeted drugs has been disappointing. Cloning of this fascinating molecule [1] raised great hopes, but it did not survive a test at the patient’s bed, so enthusiasm by many major pharmaceutical companies cooled down. As is often the case in science, as well as in drug discovery and development, projects that seemed to be dead ends unexpectedly revealed unanticipated opportunities simply by changing the strategy. A reinterpretation of the clinical use of the P2X7R might well fit in this case: it might be no longer a therapeutic target (at least until we understand in more detail its mechanistic participation in disease pathogenesis), but rather a disease marker. This new awareness may herald a second life for the P2X7R in diagnostic medicine.

## Figures and Tables

**Figure 1 ijms-24-09465-f001:**
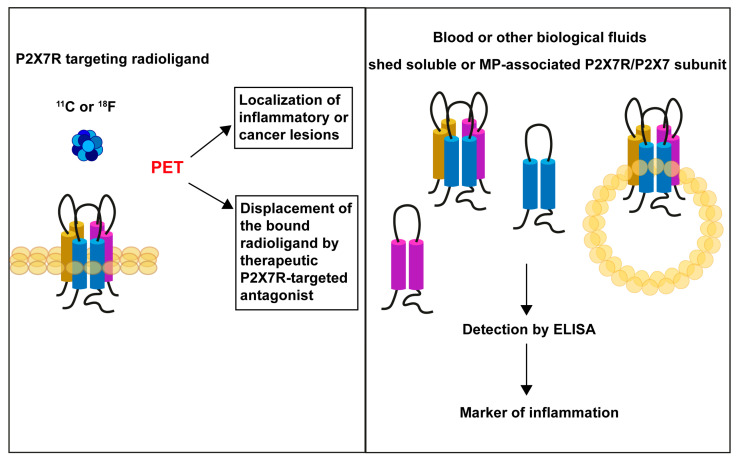
Prospective applications of the P2X7R in clinical diagnosis.

**Table 1 ijms-24-09465-t001:** P2X7R PET ligands tested in human studies.

Compounds	Ki/IC_50_ Values	In Vitro Models	Refs.
[^11^C]-GSK1482160	Ki = 5.14 ± 0.85 nM	HEK293-hP2X7R	[26]
[^11^C]-SMW139	IC_50_ = 24.5 ± 5.5 nM	1321N1-hP2X7R	[33]
[^11^C]-JNJ-54173717	IC_50_ = 4.2 nM	hP2X7R	[27]
[^18^F]-JNJ-64413739	IC_50_ = 1.0 nM	1321N1-hP2X7R	[31]

hP2X7R, human P2X7 receptor; HEK293, human embryonic kidney cells; 1321N1, human astrocyte cell line; HEK293-P2X7R, P2X7R-transfected HEK293 cells; 1321N1-P2X7R, P2X7R-transfected 1321N1 cells.

**Table 2 ijms-24-09465-t002:** Shed P2X7R blood levels (ng/L).

Sources	Healthy Subjects	Diseased Subjects
Giuliani et al. [21]	40.97 ± 3.82	204 ± 30.94 (CRP > 3 mg/L)
Conte et al. [22]	190 ± 23.9	242 ± 39.2 (epilepsy)
Garcia-Villalba et al. [23]	8–12	200 (COVID-19)

Shed P2X7R concentration from ref. [23] was inferred from data reported in Figures 1 and 2.

## Data Availability

Not applicable.

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
