# Peer review of "The Coming of Age of the P2X7 Receptor in Diagnostic Medicine"

_ijms, 2023, doi:10.3390/ijms24119465_

Round 1
Reviewer 1 Report
This review is timely, interesting and enjoyable to read and it highlights a new area of active research, namely the potential of P2X7R as a diagnostic marker of inflammatory disease and cancer. The section on the use of radioligands targeting P2X7 combined with PET is clear, although not sure of the implications surrounding the possible detection of the loss of function SNP (line 141) and why this is a caveat.
The section on shed P2X7 as a circulating marker of inflammation is more difficult to follow. The nature of the shed P2X7 is addressed at several points (starting with introduction and fig 1) and then later (section 3), where the different statements take the reader in slightly different directions. For example, on line 261 it states that only 20% of P2X7 immunoreactivity in blood is in MP fraction and then in the subsequent paragraph it states that it is likely that MP-associated P2X7R contributes to the overall P2X7R blood levels and is the main route of release.
I suggest that the issue of the nature of this shed receptor is dealt earlier on in section 3 because it is difficult to think about what the results mean without a clearer picture of the likely mechanisms contributing to the elevated levels of P2X7. The authors clearly want to remain open minded as to the nature of the shed receptor (is it a monomer, trimer, fragment etc). I found this somewhat distracting. With the available evidence so far is it not most likely that shed P2X7 is the receptor complex (trimeric) within extracellular microvesicles/MPs?
The results presented in table 2 highlight the large discrepancy in measurements of P2X7. Perhaps include more about the methodologies used in the different studies.
The quality of the English language is generally excellent with the exception of a few connecting words which aren't used in the typical way e.g. line 100, 'the search for useful targets', rather than 'of useful targets'.
Author Response
This review is timely, interesting and enjoyable to read and it highlights a new area of active research, namely the potential of P2X7R as a diagnostic marker of inflammatory disease and cancer. The section on the use of radioligands targeting P2X7 combined with PET is clear, although not sure of the implications surrounding the possible detection of the loss of function SNP (line 141) and why this is a caveat.
We sincerely thanks the Reviewer for his/her kind words. We also don't fully understand why the loss of function rs3751143 SNP might affect radioligand binding, but since this caveat was raised by Van Weehaeghe et al., we thought it fair to mention it.
The section on shed P2X7 as a circulating marker of inflammation is more difficult to follow. The nature of the shed P2X7 is addressed at several points (starting with introduction and fig 1) and then later (section 3), where the different statements take the reader in slightly different directions. For example, on line 261 it states that only 20% of P2X7 immunoreactivity in blood is in MP fraction and then in the subsequent paragraph it states that it is likely that MP-associated P2X7R contributes to the overall P2X7R blood levels and is the main route of release.
We did our best to make our reasoning smoother and easier to understand also to the lay reader.
We simplied the sentence at line 229, then we anticipated some cautionary considerations already at lines 235-239. Additionally, the nature of the shed receptor is dealt with at lines 257-261. We also apologize for the confusion arising from our statements ("about 20% of immunoreactive sP2X7R in the blood was associated to the MP fraction" and "MP-associated P2X7R contributes to the overall P2X7R blood levels"), and have included a statement that:"We think that this might be an underestimation of the actual contribution of MPs to P2X7R blood levels since MPs are very fragile, and a substantial amount might break during the isolation procedure", lines 271-273).
I suggest that the issue of the nature of this shed receptor is dealt earlier on in section 3 because it is difficult to think about what the results mean without a clearer picture of the likely mechanisms contributing to the elevated levels of P2X7. The authors clearly want to remain open minded as to the nature of the shed receptor (is it a monomer, trimer, fragment etc). I found this somewhat distracting. With the available evidence so far is it not most likely that shed P2X7 is the receptor complex (trimeric) within extracellular microvesicles/MPs?
We have included several statements aimed at clarifying this not-easy-to-follow section of the text (see lines 235-239, 257-262, 271-273).
The results presented in table 2 highlight the large discrepancy in measurements of P2X7. Perhaps include more about the methodologies used in the different studies.
We clarified that the ELISA kit used by the different groups was produced by the same manufacturer, and that two groups used plasma while the third one used both plasma and serum (lines 257-262).
The quality of the English language is generally excellent with the exception of a few connecting words which aren't used in the typical way e.g. line 100, 'the search for useful targets', rather than 'of useful targets'.
We revised the English language amending some unusual expressions. Apologies.
Reviewer 2 Report
attached

Author Response
We sincerely thank the Reviewer for his/her thoughtfull revision of our maanuscript.
His/her suggestions were incorporated.
Line 93-94. Suggest edit from of neurodegenerative diseases to ‘to understand the neurochemical basesof psychiatric disorders’
Modified.
Line 139. Suggest adding ‘ex vivo brain slices’
Modified
Line 167 and 200. Suggest the in-text references to be in chronological order.
Modified
Reviewer 3 Report
The topic of this review is interesting. The manuscript is clear and well-written. I really appreciate this paper and recommend the paper to be published after minor revisions.
(1) Changing the "clinical medicine" to "diagnostic medicine" in the title should be more consistent with the topic of this review.
(2) Line 177, “The Authors” should be “The authors”.
No comments
Author Response
We thank the Reviewer for his/her helpful comments.
All suggestions have been incorporated.
1) Changing the "clinical medicine" to "diagnostic medicine" in the title should be more consistent with the topic of this review.
Done.
2) Line 177, “The Authors” should be “The authors”.
Done